# Real-World Management of High Stool Output in Patients with Short Bowel Syndrome: An International Multicenter Survey

**DOI:** 10.3390/nu15122763

**Published:** 2023-06-16

**Authors:** Narisorn Lakananurak, Elizabeth Wall, Hilary Catron, Adela Delgado, Sophie Greif, Jean Herlitz, Lisa Moccia, David Mercer, Tim Vanuytsel, Vanessa Kumpf, Mark Berner-Hansen, Leah Gramlich

**Affiliations:** 1Division of Clinical Nutrition, Department of Medicine, Faculty of Medicine, Chulalongkorn University, King Chulalongkorn Memorial Hospital, Bangkok 10330, Thailand; 2Clinical Nutrition, University of Chicago Medicine, Chicago, IL 60637, USA; 3College of Allied Health, University of Nebraska Medical Center, Omaha, NE 68198, USA; 4Royal Alexandra Hospital, Alberta Health Services, Edmonton, AB T5H 3V9, Canada; 5Department of Hepatology and Gastroenterology, Charité—Universitätsmedizin Berlin, Corporate Member of Freie Universität Berlin, 10117 Berlin, Germany; 6Center for Human Nutrition, Cleveland Clinic, Cleveland, OH 44195, USA; moccial@ccf.org; 7Department of Surgery, University of Nebraska Medical Center, Omaha, NE 68198, USA; 8Gastroenterology and Hepatology, Katholieke Universiteit Leuven, 3000 Leuven, Belgium; 9Center for Human Nutrition, Vanderbilt University Medical Center, Nashville, TN 37232, USA; vanessa.kumpf@vumc.org; 10Digestive Disease Center K, Bispebjerg University Hospital of Copenhagen, 2400 Copenhagen, Denmark; mark.berner-hansen@regionh.dk; 11Zealand Pharma, 2860 Soeborg, Denmark; 12Department of Medicine, Faculty of Medicine and Dentistry, University of Alberta, Edmonton, AB T5B 4E4, Canada

**Keywords:** high stool output, short bowel syndrome, intestinal failure, dietary management, antimotility medication, antisecretory medication

## Abstract

Background: International practice guidelines for high-stool-output (HSO) management in short bowel syndrome (SBS) are available, but data on implementation are lacking. This study describes the approach used to manage HSO in SBS patients across different global regions. Methods: This is an international multicenter study evaluating medical management of HSO in SBS patients using a questionnaire survey. Thirty-three intestinal-failure centers were invited to complete the survey as one multidisciplinary team. Results: Survey response rate was 91%. Dietary recommendations varied based on anatomy and geographic region. For patients without colon-in-continuity (CiC), clinical practices were generally consistent with ESPEN guidelines, including separation of fluid from solid food (90%), a high-sodium diet (90%), and a low-simple-sugar diet (75%). For CiC patients, practices less closely followed guidelines, such as a low-fat diet (35%) or a high-sodium diet (50%). First-line antimotility and antisecretory medications were loperamide and proton-pump inhibitors. Other therapeutic agents (e.g., pancreatic enzymes and bile acid binders) were utilized in real-world practices, and usage varied based on intestinal anatomy. Conclusion: Expert centers largely followed published HSO-management guidelines for SBS patients without CiC, but clinical practices deviated substantially for CiC patients. Determining the reasons for this discrepancy might inform future development of practice guidelines.

## 1. Introduction

Short bowel syndrome (SBS) is a malabsorptive state resulting from extensive bowel resection or congenital diseases of the small intestine. In adults, SBS is generally defined as a clinical condition associated with remaining small bowel of less than 200 cm [1,2,3]. The overall prevalence of SBS has increased from 0.3–12 cases per million population in 1994 [4] to 3–66 cases per million population in 2010 [5] and is estimated to be even higher today due to improved outcomes achieved by multidisciplinary nutrition support teams. SBS is associated with serious and life-threatening complications, such as chronic kidney disease, catheter-related bloodstream infection, and impaired quality of life (QoL) [6,7]. Healthcare burden related to SBS is also rising, with SBS-related hospitalizations increased by 55% from 2005 to 2014, resulting in an average length of stay of 14.7 days, with an average hospital cost of USD 34,130 [8].

High stool output (HSO) is one of the most common and debilitating clinical manifestations of SBS. It generally occurs in all patients during the immediate post-operative period or hypersecretory phase and gradually improves with intestinal adaptation [3,9]. A multinational questionnaire survey for 181 patients with SBS receiving parenteral support found that diarrhea was the second-most common symptom reported in 72% of patients [10]. HSO can lead to life-threatening complications, such as severe dehydration and electrolyte disturbances, and can negatively impact QoL [3,10]. Therefore, proper management of this symptom is crucial. Multiple factors usually contribute to HSO in patients with SBS, including maldigestion and malabsorption, intestinal hypermotility, gastric acid hypersecretion, bile acid diarrhea, and small-intestinal bacterial overgrowth [11]. Hence, multimodal individualized therapy is required to decrease stool output, including dietary and oral fluid modifications, antimotility medications, antisecretory medications, and, more recently, glucagon-like peptide 2 (GLP-2) analogues.

Although international guidelines, such as the European Society for Clinical Nutrition and Metabolism (ESPEN) guidelines, were recently published, most recommendations for management of HSO in patients with SBS are based on limited or poor-quality evidence and expert opinion [1]. Real-world experience describing how clinicians incorporate guidelines into practice is of value. The aim of this study is to explore real-world practices for managing HSO among SBS experts, utilizing an international-multicenter-questionnaire-survey approach.

## 2. Materials and Methods

### 2.1. Questionnaire Survey

The SBS Management Working Group (MWG) was founded in 2020 in order to identify gaps in SBS intestinal-failure (IF) management and to find the opportunity to enhance SBS IF treatment. SBS experts across different disciplines and different global regions (North America, Europe, and Asia) were invited to join the SBS MWG, including the disciplines of gastroenterologist, surgeon, dietitian, nurse, pharmacist, and nutritionist. Using the nominal group technique for literature review, the most relevant research literature and international guidelines that were clinically useful were identified and reviewed by all members of the MWG. The gaps determined by the group were then prioritized, and a questionnaire was developed to address these gaps. The methods of this process have been described in detail previously [12,13]. HSO was considered one of the major problems in patients with SBS. The survey questionnaire included 26 questions addressing nutrition, fluid, and medication (excluding GLP-2 analogues) management of HSO in patients with SBS (Appendix A), which is the focus of the current analysis.

### 2.2. Participating Centers and Data Collection

Thirty-three international IF centers participating in the Efficacy And Safety Evaluation (EASE) phase 3 clinical trials of glepaglutide, a long-acting GLP-2 analogue in development for the treatment of SBS [14], were asked to participate in this study. The IF centers comprised of 14 IF centers located in North America (US 11 and Canada 3) and 19 IF centers located in Europe (Germany 5, UK 5, Poland 3, Denmark 2, France 2, Belgium 1, and The Netherlands 1).

The online survey tool, SurveyMonkey, was utilized to create the online questionnaire. The questionnaire was internally validated, and then a link was sent directly to the participating IF centers in January 2021. The survey questions were answered by each site as one multidisciplinary team. The questions were focused on adults (older than 18) with a diagnosis of SBS. The survey responses from the center were gathered in an anonymous fashion.

### 2.3. Ethical Statements

This study was approved by the research ethics committee of the University of Alberta, Edmonton, Alberta, Canada (Pro00107599). The data collection was based on anonymized information taken from each site. Every method was used in compliance with institutional policies and rules.

### 2.4. Statistical Analysis

Each question underwent a descriptive analysis that included tabular and graphical elements. The question-specific analysis was split by region (US vs. non-US) to detect possible interactions with these factors. The analyses were augmented with summary measures as means, medians, and ranges, as well as cross-tabulations between the replies from two of the survey questions.

## 3. Results

### 3.1. Participating Centers

The survey was completed by 30 of the 33 IF centers that were invited (with a 91% response rate). US centers dominated with a 30% response rate, followed by German (16.5%), United Kingdom (16.5%), and Canadian (10%) centers. The other centers are shown in Figure 1.

### 3.2. Dietary Recommendations for Management of High Stool Output

Dietary recommendations varied across the centers and were different between patients without and with colon-in-continuity (CiC).

In patients without CiC (i.e., with a small-intestinal stoma), the most consistent dietary recommendations were separation of fluid from solid food (90%) and a high-sodium diet (90%), followed by a low-simple-sugar diet (75%). A low-oxalate diet was never recommended to this group of patients (Table 1).

Patients with CiC were usually advised to follow a low-oxalate diet (80%) in addition to a low-simple-sugar diet (80%) and separation of fluid from solid food (75%). A low-fat diet was occasionally recommended to this group of patients (35%). Oral nutritional supplements (ONS) were suggested in 35% of practices for patients without and with CiC. The addition of soluble fiber was more frequently recommended for patients with CiC compared to patients without CiC but still only in a small proportion (25% vs. 5%, respectively) (Table 1).

There was a regional variation in dietary recommendations when comparing US and non-US centers. The US centers more frequently recommended a low-simple-sugar diet than the non-US centers for both patients without and with CiC (100% vs. 30% and 100% vs. 20%, respectively). Moreover, for patients with CiC, a low-fat diet was more often suggested in the US centers compared to the non-US centers (70% vs. 30%, respectively) (Table 2).

### 3.3. Fluid Recommendations for Management of High Stool Output

All centers recommended avoiding hyperosmolar drinks, including lemonade, sweetened tea, fruit juice, and soft drinks. Approximately one-third to one-half of the centers recommended restricting hypotonic fluid, such as coffee, alcohol, water, and tea (Figure 2).

Homemade oral rehydration solutions (ORS) were used in most centers (83%), followed by commercial ORS (76%) and sports drinks (45%) (Figure 2). The most common barrier reported for ORS use was taste fatigue (70%), followed by cost (30%) and gastrointestinal symptoms (e.g., nausea, vomiting, and abdominal pain; 20%). Reasons supporting ORS use were its efficacy (i.e., decreased stool output; 60%) and improved sense of hydration (60%).

Intravenous fluid used for rehydration included normal saline (i.e., 0.9% NaCl) in 50% of practices, followed by a balanced crystalloid solution (i.e., ringer lactate solution; 10%) and compounded solutions (10%).

### 3.4. Medications for Management of High Stool Output

#### 3.4.1. Antimotility Medications

Loperamide was most frequently used (90%), followed by codeine (20%), tincture of opium (10%), and diphenoxylate/atropine (0%). Loperamide was also the first-line antimotility medication reported in all centers (Table 3). Seventy percent of patients with SBS used more than one antimotility medication. Two-thirds of the centers (66%) reported that the strategies used to select antimotility medications did not differ based on age over/under 75.

The median (range) initial dose and the maximum dose of the two most commonly used antimotility medications were as follows: loperamide initial dose of 8 (2–24) mg/day and maximum dose of 28 (4–64) mg/day and codeine initial dose of 60 (30–120) mg/day and maximum dose of 240 (60–480) mg/day (Table 3). The main factors identified for increasing the antimotility dose were increased stool output (83%), increased stool frequency (79%), and worsening liquid stool consistency (66%).

#### 3.4.2. Antisecretory Medications

Proton-pump inhibitors (PPIs) were the most frequently used antisecretory medication (80%), followed by oral histamine H2-receptor antagonists (H2RA; 10%) and somatostatin analogues (10%). PPIs were also part of the first-line antisecretory medications, as reported by most of the centers (85%) (Table 4).

The median (range) initial and maximum doses of the most common PPI, pantoprazole, were 40 (20–80) mg/day and 80 (40–160) mg/day, respectively. The median (range) initial and maximal doses of the most common oral H2RA, famotidine, were both 40 (20–80) mg/day. Lastly, the initial and highest doses of the somatostatin analogue, octreotide, were 300 (150–600) µg/day and 600 (300–1500) µg/day, respectively (Table 4).

Most of the centers tended to continue PPIs or H2RA agents indefinitely (70% and 55%, respectively) while a minority (30% and 45%) of the centers reported stopping after 6 months.

#### 3.4.3. Other Therapeutic Agents

For patients without CiC, pancreatic enzyme supplementation and fiber were used in 20% and 10% of practices, respectively. For patients with CiC, bile acid binders, fiber, and pancreatic enzyme supplementation were occasionally used in clinical practices (30%, 20%, and 20%, respectively). Probiotics were seldom prescribed for this group of patients (10%) (Table 5).

## 4. Discussion

HSO is very common and associated with poor outcomes and low QoL in patients with SBS. In this paper, we report real-world data regarding management of HSO from expert centers around the globe.

We found variations in dietary recommendations for management of HSO in patients with SBS. For patients without CiC, most centers provided dietary recommendations consistent with the recent ESPEN guidelines and other published recommendations [1,9,15]. In contrast, not all suggested guidelines were followed for patients with CiC. For example, a high-sodium diet was not frequently recommended, perhaps because clinicians felt the absorptive capacity of the colon rendered this recommendation less critical. Recommending a low-fat diet for patients with CiC may serve to reduce steatorrhea, the risk of nephrolithiasis from calcium oxalate stones, and divalent cation (e.g., magnesium, calcium, zinc, and copper) loss in the stool [1,9]. While there are published data suggesting that a lower-fat and higher-carbohydrate diet can reduce fecal energy loss, increase overall absolute energy absorption in SBS patients with CiC [16], and promote increased energy absorption from short-chain fatty acids (SCFAs) [17], a low-fat diet was suggested in only 35% of practices in our study. This discrepancy might represent a knowledge gap or differing perspectives on what constitutes a low-fat diet (versus avoidance of a high-fat diet). Further, it could also represent the practical experience that a low-fat diet provides no overall benefit. For instance, it is observed that a low-fat diet may reduce overall intake given that it is often less palatable and less energy-dense and has lower provision of essential fatty acids and fat-soluble vitamins [1]. Previous literature showed that the addition of soluble fiber may help reduce total stool weight by 200 g/day and increase energy absorption by increased SCFAs [9,18]. However, the ESPEN guidelines do not recommend the addition of soluble fiber to increase overall energy absorption [1]. The role of fiber in patients with SBS is still controversial. It may be effective for some groups of patients or only by some types of soluble fiber. Moreover, its use can be limited by side effects, including bloating and flatulence. This study revealed that the addition of soluble fiber was recommended to some patients with SBS in real-world practices, especially to patients with CiC. Further study regarding the appropriate type of soluble fiber and patients’ characteristics is warranted.

Regional variations regarding dietary recommendations are identified in this study. This may be due to cultural variations between regions and highlights the need for individualization of dietary education. All recommendations should be tailored to each patient based on regional, cultural, and personal dietary patterns with the goals of patient understanding of the relationship between their therapeutic diet and optimization of health and hydration. A specialized and experienced dietitian is required for this level of patient education.

All centers recommended restricting oral hyperosmolar fluid, which is consistent with available guidelines [1]. Restriction of hypoosmolar fluids was recommended by half of the centers and may be based on individual patient tolerance. While patients with CiC may tolerate hyperosmolar fluid, its restriction is more important for those without CiC [9]. Regarding ORS, homemade ORS were more frequently used than commercial ORS in real-world practices. This type of ORS offers the ability to modify the recipe according to a patient’s preference, hopefully improving compliance through better palatability. Taste is a very important factor for ORS use and taste fatigue is the most important barrier found in our study. Sports drinks were recommended in almost half of the centers, even though they generally contain relatively high sugar and low sodium, and may be less appropriate for patients with SBS [9]. This may be another knowledge gap for HSO management; however, the survey did not account for centers that recommended low-sugar sports drinks or modified sports drinks in which sodium is added to make its content more suitable for the patients with SBS.

The first-line and most frequently used antimotility medication reported was loperamide. Guidelines support the use of this medication as first-line because of its efficacy to alleviate diarrhea with minimal central-nervous-system (CNS) side effects due to low oral absorption and limited ability to cross the blood–brain barrier [1]. The initial dose of 8 mg daily reported in this study is in concordance with standard recommendations. The maximal median dose of 28 mg daily and highest dose of 64 mg daily reported in this study are higher than the recommended label dose (16 mg/day) but consistent with guidelines, suggesting higher doses are effective and well tolerated in patients with SBS [11]. While assessment of tolerance to higher doses of loperamide is warranted, our findings may support the use of higher maximal doses of loperamide. Codeine was the second-most commonly used antimotility medication reported in this study, with its effectiveness offset by the increased risk of CNS side effects and potential for abuse [11]. Interestingly, the median use of diphenoxylate/atropine across all centers was reported at 0%. The unavailability of diphenoxylate/atropine in some European countries may account for this result. However, it may be ineffective to control HSO in clinical practices. Moreover, a recent systematic review showed efficacy to manage HSO with loperamide and codeine but not diphenoxylate/atropine [19].

The first-line and most frequently used antisecretory medication reported was PPIs. Extensive small-bowel resection leads to the disruption of gastro-intestinal neuroendocrine mechanisms, such as the loss of inhibitory hormones produced by the small bowel, and results in gastric acid hypersecretion [20]. The volume of gastric acid hypersecretion contributes to the total stool output, and the acidic environment can cause fat malabsorption due to interference with lipase activity. The effect may last 6–12 months postoperatively [21]. PPIs are recommended to treat the gastric acid hypersecretion [22]. It is unclear how long it is beneficial to continue use following surgery. This study found that most centers continued use indefinitely, though a minority of centers stopped PPI use after 6 months.

Alternative therapeutic agents used to manage HSO in patients with SBS were first explored in this study. Pancreatic enzyme supplementation was occasionally used in both patients without and with CiC. Rapid bowel transit in SBS may result in impaired mixing of food and endogenous pancreatic juice, and it is thought that exogenous pancreatic enzymes may help to digest and absorb nutrients in this situation, but evidence is lacking. For patients with intact colons, bile acid binders were occasionally used to treat a component of bile acid diarrhea, although SBS patients with significant ileal resection are likely to have a decreased total bile salt pool [23,24]. Soluble fiber supplements were used in some patients and may benefit patients with CiC since colonic bacteria will ferment it into SCFAs, which can facilitate fluid absorption, although studies are lacking [9].

A lot of variability, even across expert centers, was observed in this study. This may relate to the variability in SBS presentations and causes, variation in access to medications, and variation in experience. This observation has not been previously reported. It is also clear that patients with SBS and HSO require complex interventions from both diet and medication perspectives and that multidisciplinary expertise and teamwork is required to support this care.

The results of this study are limited by the generic nature of the questionnaire survey design. While the response rate is high and represents a diverse geographical area within Europe and North America, it was only distributed to IF centers participating in the EASE trial. Nevertheless, to the best of our knowledge and review of the literature, this is the first study reporting real-world experience regarding management of HSO within IF centers. This data can potentially serve to guide clinicians in the management of HSO in patients with SBS.

## 5. Conclusions

This study revealed that while expert centers generally followed published guidelines for the management of HSO in patients without CiC, this was much less common in patients with CiC. Dietary management strategies varied based on bowel anatomy and were different between centers and geographical regions. Loperamide and PPIs were commonly utilized by IF centers for medical management of HSO in patients with SBS. Our findings identified the use of loperamide doses exceeding guideline recommendations in real-world practices. Other therapeutic agents used by IF centers to manage HSO included pancreatic enzymes and bile acid binders, despite limited availability of supporting evidence. These data suggest areas of significant discrepancy between what is published in guidelines and what is actually used in practices. This could represent gaps in knowledge, but could equally point to a lack of validity of current guidelines. Because current guidelines are presently so dependent on expert opinion, this report of actual clinical practices might inform the development of future recommendations.

## Figures and Tables

**Figure 1 nutrients-15-02763-f001:**
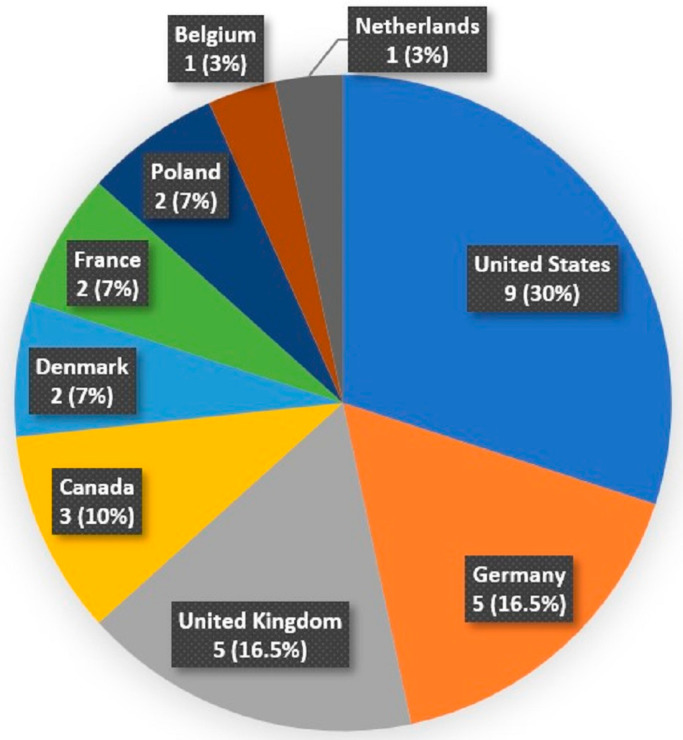
Geographic location of the participating intestinal-failure centers.

**Figure 2 nutrients-15-02763-f002:**
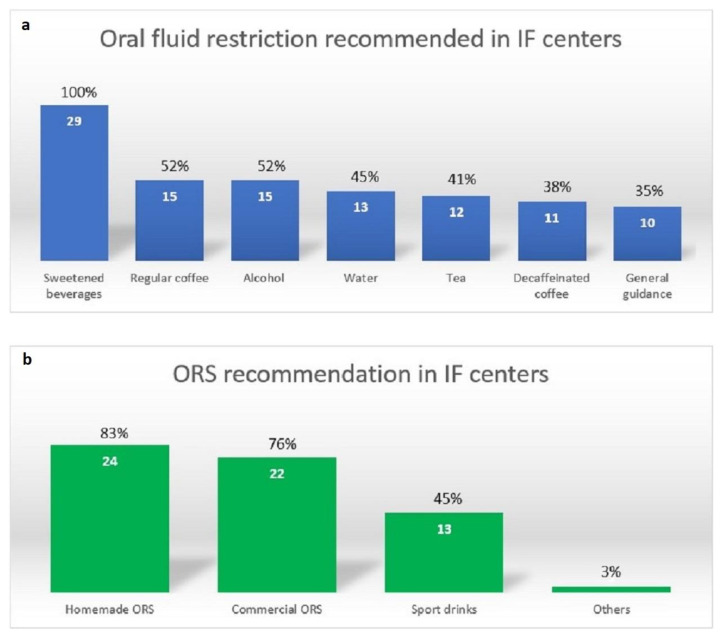
(**a**) Oral fluid restriction and (**b**) type of ORS recommendation for patients with SBS (*n* = 29). Abbreviations: ORS, oral rehydration solution; SBS, short bowel syndrome; IF, intestinal failure.

**Table 1 nutrients-15-02763-t001:** Dietary recommendations for patients with short bowel syndrome and high stool output without or with colon-in-continuity.

Dietary Recommendations	Without Colon-in-Continuity Median (Range, %)	With Colon-in-Continuity Median (Range, %)
A low-simple-sugar diet	75 (0–100)	80 (0–100)
A low-oxalate diet	0 (0–100)	80 (0–100)
Separation of fluid from solid food	90 (0–100)	75 (0–100)
A high-sodium diet	90 (0–100)	50 (0–100)
A low-fat diet	10 (0–60)	35 (0–100)
Oral nutritional supplements	35 (0–100)	35 (0–100)
Addition of soluble fiber	5 (0–90)	25 (0–90)

**Table 2 nutrients-15-02763-t002:** Dietary recommendations for patients with short bowel syndrome and high stool output without or with colon-in-continuity, showing a comparison between US (*n* = 9) and non-US (*n* = 21) centers.

Dietary Recommendations	Without Colon-in-Continuity		With Colon-in-Continuity	
	US Median (Range, %)	Non-US Median (Range, %)	US Median (Range, %)	Non-US Median (Range, %)
A low-simple-sugar diet	100 (90–100)	20 (0–100)	100 (10–100)	30 (0–100)
A low-oxalate diet	0 (0–100)	0 (0–100)	50 (0–100)	90 (0–100)
Separation of fluid from solid food	100 (10–100)	90 (0–100)	100 (10–100)	70 (0–100)
A high-sodium diet	100 (0–100)	80 (0–100)	60 (0–100)	40 (0–100)
A low-fat diet	10 (0–60)	20 (0–60)	70 (0–100)	30 (0–100)
Oral nutritional supplements	10 (0–100)	50 (0–100)	30 (0–100)	40 (0–100)
Addition of soluble fiber	20 (0–70)	0 (0–90)	40 (0–70)	20 (0–90)

**Table 3 nutrients-15-02763-t003:** Antimotility medications used for patients with short bowel syndrome.

Medications	Usage Median (Range, %)	Initial Dose Median (Range, mg/day)	Maximum Dose Median (Range, mg/day)	Dose Recommendation per Label (mg/day)
Loperamide (First-line)	90 (40–100)	8 (2–24)	28 (4–64)	2–6 mg QID; maximum daily dose, 16 mg
Codeine	20 (0–100)	60 (30–120)	240 (60–480)	15–60 mg QID
Tincture of opium	10 (0–100)	8 (12–17.5)	35 (24–50)	0.3–1 mL QID
Diphenoxylate and atropine	0 (0–100)	10 (7.5–20)	20 (10–40)	2.5–7.5 mg QID; maximum daily dose, 20–25 mg

Abbreviations: mg, milligram; QID, four times a day.

**Table 4 nutrients-15-02763-t004:** Antisecretory medications used for patients with short bowel syndrome.

Medication Class	Usage Median (Range, %)	Medications	Initial Dose Median (Range, mg/day)	Maximum Dose Median (Range, mg/day)	Dose Recommendation per Label (mg/day)
Proton-pump inhibitor (first-line)	80 (0–100)	Pantoprazole	40 (20–80)	80 (40–160)	20–40 mg BID
		Omeprazole	40 (20–80)	80	20–40 mg BID
		Esomeprazole	80	80	20–40 mg BID
		Lansoprazole	60	60	15–30 BID
H2RA (oral)	10 (0–30)	Famotidine	40 (20–80)	40 (20–80)	20–40 mg BID
		Ranitidine	300	300	150–300 mg BID
		Nizatidine	150	300	150–300 mg BID
Somatostatin analogue	10 (0–50)	Octreotide	300 (150–600) μg	600 (300–1500) μg	50–250 μg SC TID or QID
H2RA (added to PS)	0 (0–90)	Famotidine	40 (20–80)	40 (20–80)	20–40 mg BID

Abbreviations: mg, milligram; H2RA, histamine 2-receptor antagonists; BID, two times a day; TID, three times a day; QID, four times a day; PS, parenteral support.

**Table 5 nutrients-15-02763-t005:** Other therapeutic agents used in patients with short bowel syndrome without or with colon-in-continuity.

Therapeutic Agents	Without Colon-in-Continuity Median (Range, %)	With Colon-in-Continuity Median (Range, %)
Bile acid binders	0 (0–100)	30 (0–100)
Fiber (e.g., psyllium)	10 (0–90)	20 (0–90)
Pancreatic enzymes	20 (0–80)	20 (0–80)
Probiotics	0 (0–60)	10 (0–100)

## Data Availability

Data is unavailable due to privacy concern.

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
