# Peer review of "Real-World Management of High Stool Output in Patients with Short Bowel Syndrome: An International Multicenter Survey"

_nutrients, 2023, doi:10.3390/nu15122763_

Round 1

Reviewer 1 Report

This is an article on the clinical management of HSP in patients with short bowel syndrome. It summarizes the management experience of multiple intestinal failure centers worldwide through a questionnaire, and provides specific recommendations on some details such as the dosage of PPI. But compared to guidelines for short bowel syndrome, there are no new highlights or findings. I think the author can add some new drugs, such as the experience of GLP-2 analogues and comparison of patient data.

This is an article on the clinical management of HSP in patients with short bowel syndrome. It summarizes the management experience of multiple intestinal failure centers worldwide through a questionnaire, and provides specific recommendations on some details such as the dosage of PPI. But compared to guidelines for short bowel syndrome, there are no new highlights or findings. I think the author can add some new drugs, such as the experience of GLP-2 analogues and comparison of patient data.

Author Response

Cover letter for revision of the manuscript

June 7, 2023

Dear Editor,

            Thank you for giving me the opportunity to revise and resubmit the manuscript entitled " Real-World Management of High Stool Output in Patients with Short Bowel Syndrome: An International Multicenter Survey" I appreciate the careful review and constructive suggestions provided by each reviewer. The manuscript has certainly benefited from these insightful revision suggestions. I look forward to hearing from you regarding our submission and to respond to any further questions and comments you may have.

            Following this letter are the reviewers’ comments with our specifically responses to each suggestion, including how and where the text was modified. Changes made in the manuscript are marked using yellow highlighted text.

            All authors have reviewed the adapted manuscript and agree with resubmission.

Please address all correspondence concerning this manuscript to me at Narisorn.L@chula.ac.th

Sincerely,

Narisorn Lakananurak MD MSc

Division of Clinical Nutrition, Department of Medicine

Chulalongkorn University, King Chulalongkorn Memorial Hospital

Bangkok, Thailand

Leah Gramlich MD.

Division of Gastroenterology, Department of Medicine,

Faculty of Medicine and Dentistry, University of Alberta,

Edmonton, Alberta, Canada

Reviewers’ Comments

Reviewer 1

Comments to the Author

1. This is an article on the clinical management of HSO in patients with short bowel syndrome. It summarizes the management experience of multiple intestinal failure centers worldwide through a questionnaire, and provides specific recommendations on some details such as the dosage of PPI. But compared to guidelines for short bowel syndrome, there are no new highlights or findings.

Answer

             Thank you for the valuable comments. The goal of this study, however, was not to review the guidelines but to elucidate how clinicians in different global regions incorporate the guidelines into real-world practice. We agree that the intestinal failure centers followed the published guidelines for some recommendations. However, not all recommendations were followed in real-world practice. As the recommendations in the guidelines are generally based on low quality of evidence and expert opinions, the data reported in this study is of value to guide clinicians in the management of high stool output in patients with short bowel syndrome (SBS) and may also be relevant for the development of future guidelines. To the best of our knowledge, this is the first study that revealed real-world management of high stool output in patients with SBS. Additionally, the discrepancies between guidelines and real-world practice shown in this study can be used to guide future studies to evaluate underlying reasons and build a base for potential guideline modifications.

            Examples of novel and important data found in our study are shown as follows and we have emphasized and added them in the manuscript according to your suggestion:

- The ESPEN guidelines [1] recommend a low-fat diet in patients with colon-in-continuity to reduce risk of nephrolithiasis, mineral losses, and increase energy absorption. However, this study showed that a low-fat diet was not frequently recommended in real-world practice. This discrepancy might represent a knowledge gap, or differing perspectives on what constitutes a low-fat diet (versus avoidance of a high-fat diet). Further, it could also represent the practical experience that a low-fat diet provides no overall benefit.  For instance, it is observed that a low-fat diet may reduce overall intake given that it is often less palatable and less energy dense and has lower provision of essential fatty acids and the fat-soluble vitamins. (page 7, line 206-218) This data can guide the future study to evaluate the necessity of a low-fat diet in SBS patients with colon-in-continuity.

- Previous literature showed that addition of soluble fiber may help reduce total stool weight by 200 g/day and increase energy absorption by increased SCFAs. [2,3] However, the ESPEN guidelines do not recommend addition of soluble fiber to increase overall energy absorption. [1] The role of fiber in patients with SBS is still controversial. It may be effective for some groups of patients or only by some types of soluble fiber. Moreover, its use can be limited by side effects including bloating and flatulence. This study revealed that addition of soluble fiber was recommended in real-world practice, especially in patients with colon-in-continuity. Further study regarding the appropriate type of soluble fiber and patients’ characteristics is warranted.  We have added this on page 7, line 218-227.

- Homemade oral rehydration solution (ORS) was more frequently used than commercial ORS in real-world practice. This type of ORS offers the ability to modify the recipe according to a patient’s preference, hopefully improving compliance through better palatability.  Taste is a very important factor for ORS use and taste fatigue is the most important barrier found in our study. (page 8, line 239-243) This data can guide clinicians how to choose and modify ORS to achieve maximal patient’s compliance.

- Loperamide was the most commonly used and first line antimotility medication. Though higher dose is recommended in the guidelines, a much higher maximum dose of loperamide was utilized in real-world practice (up to 64 mg/day). While assessment of tolerance to higher doses of loperamide is warranted, our findings may support the use of higher maximal doses of loperamide. (page 8, line 254-258)

- Although diphenoxylate/atropine is recommended to reduce intestinal motility, our study showed that it was rarely prescribed in real-world practice. This may represent the unavailability of diphenoxylate/atropine in some European countries. However, it may be ineffective to control stool output in clinical practice. Moreover, a recent systematic review showed efficacy to manage high stool output with loperamide and codeine, but not diphenoxylate/atropine. [4] (page 8, line 260-265)

- The evidence regarding usage of alternative therapeutic agents, such as pancreatic enzymes, bile acid binders, and probiotics, is lacking. The usage of these agents in real-world practice was first explored in our study. We found that pancreatic enzyme supplementation was occasionally used in both patients without and with colon-in-continuity while bile acid binders and fiber were more frequently used in patients with colon-in-continuity. (page 8, line 275-284) Clinicians can use our data to guide how to prescribe these medications and agents for patients with SBS.

- A lot of variability even across expert centers was first observed in this study. This may relate to the variability in SBS presentations and causes, variation in access to medications and variation in experience. This observation has not been previously reported. It is also clear that patients with SBS and HSO require complex interventions from both diet and medication perspectives and that multidisciplinary expertise and teamwork is required to support this care. We have added this statement on page 9, line 286-291.

2. I think the author can add some new drugs, such as the experience of GLP-2 analogues and comparison of patient data.

Answer: Thank you very much. We agree with your comment that the experience of using GLP-2 analogues is very important and should be explored. Therefore, we included 16 questions regarding the use of GLP-2 analogues in our survey. After the analysis, there are a lot of interesting data regarding the use of GLP-2 analogues. However, there are also many interesting and important data regarding nutrition, fluid, and medications management of high stool output. Including all of these data in one manuscript may result in too much information and may lead to a loss of focus on each important information. Therefore, we decided to report the data regarding GLP-2 analogues separately in another manuscript.

            We have compared data (e.g., US vs. non-US, patients with vs. without colon-in-continuity) and have selected important data to show in our manuscript according to your suggestion.

Reference

  1. Cuerda, C.; Pironi, L.; Arends, J.; Bozzetti, F.; Gillanders, L.; Jeppesen, P.B.; Joly, F.; Kelly, D.; Lal, S.; Staun, M.; et al. ESPEN practical guideline: Clinical nutrition in chronic intestinal failure. Clin Nutr 2021, 40, 5196-5220, doi:10.1016/j.clnu.2021.07.002.
  2. Limtrakun, N.; Lakananurak, N. Dietary Strategies for Managing Short Bowel Syndrome. Current Treatment Options in Gastroenterology 2022, 20, 376-391, doi:10.1007/s11938-022-00385-y.
  3. Hamilton, K.; Crowe, T.; Testro, A. High amylase resistant starch to decrease stool output in people with short bowel syndrome: A pilot trial. Clin Nutr ESPEN 2019, 29, 242-244, doi:10.1016/j.clnesp.2018.10.006.
  4. de Vries, F.E.E.; Reeskamp, L.F.; van Ruler, O.; van Arum, I.; Kuin, W.; Dijksta, G.; Haveman, J.W.; Boermeester, M.A.; Serlie, M.J. Systematic review: pharmacotherapy for high-output enterostomies or enteral fistulas. Aliment Pharmacol Ther 2017, 46, 266-273, doi:10.1111/apt.14136.

Reviewer 2 Report

The manuscript titled "Real-World Management of High Stool Output in Patients with Short Bowel Syndrome: An International Multicenter Survey" is highly relevant to the field of interest. It is well-structured and includes ample data. However, there are some minor issues that need to be addressed. Therefore, I recommend this manuscript for minor revision.

If there are 34 questions in the questionnaire model, please provide all of them in the supplementary materials. The questionnaire questions begin from number 9.

In Figure 2, please label the images as 2a and 2b

Please remove the italics from lines 127-131.

Instead of using abbreviations separately, utilize the full form of the abbreviation in all table titles.

Author Response

Cover letter for revision of the manuscript

June 7, 2023

Dear Editor,

            Thank you for giving me the opportunity to revise and resubmit the manuscript entitled " Real-World Management of High Stool Output in Patients with Short Bowel Syndrome: An International Multicenter Survey" I appreciate the careful review and constructive suggestions provided by each reviewer. The manuscript has certainly benefited from these insightful revision suggestions. I look forward to hearing from you regarding our submission and to respond to any further questions and comments you may have.

            Following this letter are the reviewers’ comments with our specifically responses to each suggestion, including how and where the text was modified. Changes made in the manuscript are marked using yellow highlighted text.

All authors have reviewed the adapted manuscript and agree with resubmission.

Please address all correspondence concerning this manuscript to me at Narisorn.L@chula.ac.th

Sincerely,

Narisorn Lakananurak MD MSc

Division of Clinical Nutrition, Department of Medicine

Chulalongkorn University, King Chulalongkorn Memorial Hospital

Bangkok, Thailand

Leah Gramlich MD.

Division of Gastroenterology, Department of Medicine,

Faculty of Medicine and Dentistry, University of Alberta,

Edmonton, Alberta, Canada

Reviewer 2

Comments to the Author

The manuscript titled "Real-World Management of High Stool Output in Patients with Short Bowel Syndrome: An International Multicenter Survey" is highly relevant to the field of interest. It is well-structured and includes ample data. However, there are some minor issues that need to be addressed. Therefore, I recommend this manuscript for minor revision.

1. If there are 34 questions in the questionnaire model, please provide all of them in the supplementary materials. The questionnaire questions begin from number 9.

Answer: Thank you for the suggestion. All 34 questions have already been included in the questionnaire model (please see supplemental material).

2. In Figure 2, please label the images as 2a and 2b

Answer: We labeled figure 2 as 2a and 2b according to your comment (page 5).

3. Please remove the italics from lines 127-131.

Answer: The italics between lines 127-131 have been removed (updated line number 132-137).

4. Instead of using abbreviations separately, utilize the full form of the abbreviation in all table titles.

Answer: The full form of the abbreviation has been used in all tables.
